# Magnetocaloric Effect for a *Q*-Clock-Type System

**DOI:** 10.3390/e27010011

**Published:** 2024-12-27

**Authors:** Michel Aguilera, Sergio Pino-Alarcón, Francisco J. Peña, Eugenio E. Vogel, Natalia Cortés, Patricio Vargas

**Affiliations:** 1Instituto de Física, Pontificia Universidad Católica de Valparaíso, Casilla 4950, Valparaíso 2373223, Chile; 2Departamento de Física, Universidad Técnica Federico Santa María, Av. España 1680, Valparaíso 2390123, Chile; sergio.pinoa@usm.cl (S.P.-A.); f.penarecabarren@gmail.com (F.J.P.); patricio.vargas@usm.cl (P.V.); 3Millennium Nucleus in NanoBioPhysics (NNBP), Av. España 1680, Valparaíso 2390123, Chile; 4Departamento de Ciencias Físicas, Universidad de La Frontera, Casilla 54-D, Temuco 4811230, Chile; eugenio.vogel@ufrontera.cl; 5Facultad de Ingeniería, Universidad Central de Chile, Santiago 8330601, Chile; 6Instituto de Alta Investigación, Universidad de Tarapacá, Casilla 7D, Arica 1000000, Chile; natalia.cortesm@usm.cl

**Keywords:** entropy, *Q*-clock, magnetocaloric

## Abstract

In this work, we study the magnetocaloric effect (MCE) in a working substance corresponding to a square lattice of spins with *Q* possible orientations, known as the “*Q*-state clock model”. When the *Q*-state clock model has Q≥5 possible configurations, it presents the famous Berezinskii–Kosterlitz–Thouless (BKT) phase associated with vortex states. We calculate the thermodynamic quantities using Monte Carlo simulations for even *Q* numbers, ranging from Q=2 to Q=8 spin orientations per site in a lattice. We use lattices of different sizes with $N=L\times L=8^{2},16^{2},32^{2},64^{2},\mathrm{and}\;128^{2}$ sites, considering free boundary conditions and an external magnetic field varying between B=0 and B=1.0 in natural units of the system. By obtaining the entropy, it is possible to quantify the MCE through an isothermal process in which the external magnetic field on the spin system is varied. In particular, we find the values of *Q* that maximize the MCE depending on the lattice size and the magnetic phase transitions linked with the process. Given the broader relevance of the *Q*-state clock model in areas such as percolation theory, neural networks, and biological systems, where multi-state interactions are essential, our study provides a robust framework in applied quantum mechanics, statistical physics, and related fields.

## 1. Introduction

Caloric phenomena form a foundational aspect of material physics, crucial in identifying viable substitutes for the toxic gases currently employed in conventional refrigeration systems [1,2]. Significant temperature variations driven by caloric processes have been observed in different materials following adiabatic changes in applied hydrostatic pressure (barocaloric effect) [3,4,5,6], mechanical stress (elastocaloric effect) [7,8,9,10,11], the electric field (electrocaloric effect) [12,13,14,15], and the external magnetic field (magnetocaloric effect, MCE) [16,17,18,19]. Although the underlying nature and physical interpretation of each phenomenon differ, they share a common objective: maximizing entropy changes in response to variations in the control parameter governing the thermodynamic process [20,21,22,23,24,25,26].

Systems exhibiting phase transitions, in particular, achieve maximum entropy changes near where these transitions are occurring [27,28,29,30]. Among these, magnetic materials stand out due to their distinct magnetic order or ground state, which is heavily influenced by the energy contributions within the material [31,32,33]. Here, we concentrate on magnetic systems that undergo phase changes with increasing temperature, specifically those that alter their magnetic phase. A notable model for exploration is the *Q*-state clock model, a discrete variant of the renowned 2D XY model [34,35,36,37], widely studied for illustrating the Berezinskii–Kosterlitz–Thouless (BKT) transition in frustrated quenched disordered phases [38,39,40]. The *Q*-state clock model serves as a classical Heisenberg spin model with pronounced planar anisotropy, useful in replicating material thermodynamics [41,42,43].

Phase transitions in the *Q*-clock state model can be identified via the maxima observed in the specific heat as a function of temperature. Each maximum signifies there is a “critical temperature”. In scenarios devoid of an external magnetic field, studies have demonstrated [44,45,46,47,48,49] that for Q≥5, the specific heat exhibits two maxima. The initial maximum pertains to the transition from a ferromagnetic phase (FP) to a BKT phase, followed by a transition from BKT to a paramagnetic disordered phase (PP) [36].

Due to these dual phase transitions, the *Q*-clock state model is particularly intriguing for caloric phenomenon studies. This model prompts several critical questions: What is the optimal value of *Q* that maximizes entropy variation and, consequently, the MCE? Which magnetic phases enhance the effect most effectively? Are the results consistent as the lattice size increases?

A major technical challenge with this model revolves around scaling accessible states, which scales approximately as $∼Q^{L^{2}}$, where N=L×L defines the lattice size with *N* number of sites. Consequently, exact computations using the canonical partition function entail substantial expenses, thus restricting exact computations to smaller lattices. Mean-field approximation and Monte Carlo simulations allow us to address large lattice size challenges. Monte Carlo simulations are often preferred for their accuracy in representing systems with intricate interactions and fluctuations, which the mean-field approach may overlook due to its averaging assumptions. Consequently, mean-field theory may not aptly describe short-range interactions like those in the systems examined herein.

Material systems where the *Q*-state clock model can be directly applied are complex due to its theoretical nature. Nonetheless, this model is useful for studying phase transition properties that occur in two-dimensional (2D) arrays [35,50]. The type of interactions between the discrete spin configurations in a 2D lattice plays a fundamental role in the *Q*-state clock model. A scenario where one could find applications for this model corresponds to artificial spin ice systems. In this case, the materials used are permalloys with square lattices [51,52], which may be in direct analogy with the *Q*-state clock model by choosing appropriate *Q* values into a Hamiltonian that includes short- and long-range interactions.

This work investigates the MCE in the *Q*-clock model, utilizing Monte Carlo simulations to derive the model’s thermodynamic properties for large lattice sizes. The analyzed *Q* values are even numbers (Q=2,4,6,8), facilitating exploration into how the spin orientation given by *Q* numbers influences the magnetocaloric effect across variations in thermal and magnetic conditions. We examine the magnetocaloric effect’s reliance on these *Q* values, focusing on entropy, specific heat, internal energy, and magnetization responses near phase transitions. This study elucidates optimal spin configurations to maximize the MCE in discrete multi-state systems.

Our work is structured as follows: Section 2 introduces the *Q*-state clock model, detailing its Hamiltonian and the relevant thermodynamic quantities. Section 3 describes the Monte Carlo simulation techniques used to derive thermodynamic properties, including detailed explanations of lattice sizes, boundary conditions, and sampling methodologies. Section 4 examines the calculated thermodynamic properties, such as specific heat, internal energy, and entropy, for various values of *Q*. Section 5 explores the magnetocaloric effect and outlines the methodology for its calculation. Section 6 presents our findings, highlighting the optimal conditions and *Q* values that maximize the effect. Finally, Section 7 summarizes the key findings of our research.

## 2. Spin Model

### Q-State Clock Model

The system under study corresponds to the *Q*-states clock model on a two-dimensional (2D) square lattice of dimensions L×L=N (with *N* the total number of sites in the lattice). The local magnetic moment or “spin” ${\vec{S}}_{i}$ at site *i* can point in any of *Q* directions in a given plane, see Figure 1. ${\vec{S}}_{i}$ is then a 2D vector, i.e., ${\vec{S}}_{i}=\left(\cos\left(\frac{2\pi }{Q}k_{i}\right),\sin\left(\frac{2\pi }{Q}k_{i}\right)\right)$, where $k_{i}=0,1,\ldots Q-1$, with equal probability and the index *i* runs over all sites of the lattice. The magnitude of ${\vec{S}}_{i}$ is chosen as the unity.

The isotropic Hamiltonian for such a system can be written as follows:(1)$$H=-\sum_{\langle i,j\rangle }J{\vec{S}}_{i}\cdot {\vec{S}}_{j}-\sum_{i}\vec{B}\cdot {\vec{S}}_{i}\;,$$
where J>0 is the ferromagnetic exchange interaction to the nearest neighbors; the sum runs over all pairs of nearest neighbors 〈i,j〉. $\vec{B}$ is an external magnetic field applied along one of the directions of the plane. We use J=1 in our calculations; thus, all quantities are delivered in exchange units.

To quantify the caloric effect, the entropy from the thermodynamics of the model must be obtained. To do that, we will analyze Monte Carlo simulations, which offer the possibility of obtaining thermal averages within a sample of the possible configurations of the system. The following section will detail the method employed, including the algorithm used in our computational calculations.

## 3. Monte Carlo Simulations

The work presented herein will focus on numerical computations based on Monte Carlo (MC) simulations. A square lattice of size L×L is selected, with free boundary conditions imposed. A site is randomly chosen for visitation, and the energy cost, Δ, associated with rotating the corresponding spin among *Q* possible states is calculated. If the energy is diminished, the change in orientation is accepted; otherwise, the spin rotation is accepted only when exp(−Δ/T)≤r, where *r* is a freshly generated random number uniformly distributed in the range [0, 1]. This procedure conforms to the conventional Metropolis algorithm. A Monte Carlo step (MCS) is achieved after N=L×L spin-rotation attempts.

For each lattice size and *Q* value, a sequence of temperatures has been established within the range [0.02, 4] in increments of 0.02 for each temperature. A total of 5τ MCSs are performed: the initial τ MCSs are utilized to equilibrate the system at a fixed temperature *T*. In contrast, the subsequent 4τ MCSs are employed to measure observables every 20 MCSs, achieving a cumulative total of $2\times 10^{5}=200.000$ measurements. From now on, $\tau =10^{6}$, unless otherwise specified for the remainder of this paper, and this choice of τ yields stable results and corresponds with the analytical expressions obtained in previous studies for smaller lattices.

### 3.1. Thermal Averages

The magnetization per site $\vec{M}$ is given by
(2)$$\vec{M}=\frac{1}{N}\sum_{i=1}^{N}{\vec{S}}_{i},$$
where ${\vec{S}}_{i}$ is the spin at site *i* at a given time, *t*, and N=L×L. In this case, $\vec{M}$ is a vector of two components $\vec{M}=\left(M_{x},M_{y}\right)$. Normally, this vector’s magnitude or absolute value is calculated as $|\vec{M}|=\sqrt{M_{x}^{2}+M_{y}^{2}}$. Then, the thermal average of $|\vec{M}|$ is $<|\vec{M}|>$ given by
(3)$$<|\vec{M}|>=\frac{1}{N_{c}}\sum_{i=1}^{N_{c}}\sqrt{M_{x}^{2}+M_{y}^{2}},$$
with $N_{c}=2\times 10^{5}$ the number of configurations used to perform thermal averages.

Energy is the main quantity the Monte Carlo method uses to reach thermal equilibrium. Therefore, after τ MCSs, the internal energy *U* can be obtained by averaging the $N_{c}=2\times 10^{5}$ values for $E_{k}$, where *k* runs over the accepted configurations after the Metropolis algorithm, namely,
(4)$$U=<E>=\frac{1}{N_{c}}\sum_{k=1}^{N_{c}}E_{k},$$
where every spin configuration is separated from the next one by 20 MCSs. The energy per site is then U/N.

The specific heat is then calculated as proportional to the fluctuations in the energy as follows (we use $k_{B}=1$ for simplicity):(5)$$C=\frac{\langle E^{2}\rangle -{\langle E\rangle }^{2}}{T^{2}},$$
(6)$$C=\frac{1}{T^{2}}\left[\left(\frac{1}{N_{c}}\sum_{k=1}^{N_{c}}E_{k}^{2}\right)-{\left(\frac{1}{N_{c}}\sum_{k=1}^{N_{c}}E_{k}\right)}^{2}\right].$$
The entropy *S* can be calculated by numerical integration of the specific heat over *T* as
(7)$$S\left(T,B\right)=S_{0}+\int \frac{C(T,B)}{T}dT.$$

Our analysis determines the entropy at zero temperature under zero and non-zero magnetic field conditions by examining the energy degeneracy inherent at (T=0). Without a magnetic field, (Q) ferromagnetic spin configurations exist, each possessing identical energies, resulting in an entropy of $S_{0}=\ln Q$. Consequently, the entropy at any temperature *T* with a zero magnetic field is given by $S\left(T,0\right)=\ln Q+\int \frac{C(T,0)}{T}dT$. Conversely, when the magnetic field (B≠0), there is a unique ground state where all the spins align uniformly with the field, leading to the constant $S_{0}$ going to zero.

### 3.2. Selection of the Lattice Size

Because of the scaling of the system’s accessible states ($∼Q^{L^{2}}=Q^{N}$), it is necessary to set a criterion for a representative lattice size in the thermodynamic limit, where the results of the caloric studies are valid and closer to reality. For this purpose, we have analyzed the convergence of the internal energy *U* for different values of the clock parameter *Q*, and the lattice size *L* with temperature *T* as the independent variable, using Monte Carlo simulations. For this purpose, we have analyzed the system’s internal energy, normalized to the number of sites, U(Q,L,T)/N (or simply U/N), as a function of *T*. The clock parameter *Q* takes the values 2, 4, 6, and 8, while the lattice sizes N=L×L are decimated, taking values *L* = 8, 16, 32, 64, and 128. In addition, we consider two values for the external magnetic field: B=0.0 and B=1.0.

The results are given in Figure 2. The left panel gives the normalized energy without an external magnetic field, while the calculations reported in the right panel correspond to an external magnetic field B=1.0. The convergence from the upper curves (smaller lattices) to the lower curves (larger curves) is notorious. However, to investigate the stability of the curves for L=128, we conduct a complementary study. For a square lattice with L×L sites, it is convenient to plot the energy per site as a function of 1/L for the scaling analysis, as we show in Figure 3 for two different temperatures: $T^{*}=0.5$ and $T^{*}=3.5$. Since the results tend to coincide and overlap in the plots, we have left out the curve for the case Q=2, which corresponds to the well-known Ising model, thus leaving the upper *Q* values in the plot.

In Figure 3, we have also included a linear fit as straight lines: solid for B=0.0 and dashed for B=1.0. This allows us to obtain the intersects for the thermodynamic limit: $U(Q,\infty ,T^{*})/N$. Then, we can calculate the percent deviation ϵ of the L=128 results with respect to these limiting values, namely,
(8)$$\epsilon =100\times \frac{U\left(Q,128,T^{*}\right)-U\left(Q,\infty ,T^{*}\right)}{U(Q,128,T^{*})}.$$

The values of this parameter are reported in a table within each panel in Figure 3, including here the case Q=2 that was not plotted.

The values reported for L=128 are near what can be expected for U/N in the thermodynamic limit. Actually, the largest deviation observed is only 1.1%. From now on, we will analyze the results for L=128 as a representative of the thermodynamic limit.

## 4. Analysis of Thermodynamic Quantities

We will begin by reviewing the thermodynamic results of the model, focusing first on the analysis of the specific heat for a 128×128 lattice. This thermodynamic quantity for a field B=0 and field B=1 is shown in Figure 4a,b, respectively, for Q=2,4,6,8. Here, we observe two peaks for Q≥6, indicative of a double-phase transition. It is, therefore, the BKT phase that appears in these cases. It is also noticeable that the critical temperature of each transition increases as *B* increases, and their associated peaks are less pronounced, as we can observe when comparing Figure 4a,b. This is because the external magnetic field favors ordered phases (FM and BKT) over disordered ones.

Following with the 128×128 lattice, its internal energy as a function of *T* is shown in Figure 5a, where we can observe that lower values are obtained when B=1.0 (blue curves) than when B=0.0 (purple curves). This is due to the Zeeman term present in the Hamiltonian given by Equation (Equation 1), which decreases the ground-state energy compared to the case in the absence of a magnetic field. Figure 5b shows the magnetization *M* of the system for Q=2,4,6, and 8 for B=1 (blue curves) and B=0 (purple curves). First, we notice that all systems start saturated (in a ferromagnetic state), and then *M* decreases as *T* increases. The cases with B=1.0 show higher magnetization than those with B=0.0 as *T* increases for all *Q* displayed. It is observed that the change in magnetization as a function of temperature happens much faster for larger *Q* values. This is a direct consequence of the degrees of freedom of the system. For example, we can think of Q=2, where two possible orientations for the spin are up and down. In terms of energy, generating a flip of such a configuration is much more difficult. Therefore, the magnetization will be higher for low *Q* than for higher values of *Q* at the same temperature.

Next, we analyze the entropy in Figure 5c. We observe that for a given *Q*, the entropy is lower when B=1 (blue curves) than for the case of B=0 (purple curves). This is consistent with the interpretation that the magnetic field tends to order the spin system. Also, it is noticeable that for more spin degrees of freedom (as *Q* increases), the entropy is higher as a function of temperature, which is consistent with the more significant number of accessible states the system acquires as *Q* increases. These curves show that the entropy fulfills $\lim_{T\to 0}S\left(T,B\neq 0\right)=0$ because, with even a very low field value, the system at low temperature tends to have only one preferred state due to the presence of the external field applied to the system. It is important to remember that in the case of a null field, the entropy at low temperature has the value of $\lim_{T\to 0}S\left(T,0\right)=\ln Q$. It should be noted that the entropy in Figure 5 is normalized to the total number of spins in the lattice, i.e., S(T,B)/N. This implies that the quantity $S_{0}/N$ is very small when the field tends to zero (and strictly zero with B≠0). In the case of the lattice size 128×128 displayed in Figure 5, this number would be given by ln(Q)/16384, in which the most significant value would be provided for Q=8, offering a correction of the order of $10^{-4}$.

## 5. Magnetocaloric Phenomena

To analyze the magnetocaloric effect, we start by treating the total entropy of a material as the sum of three entropies: electronic $S_{e}$, magnetic $S_{m}$, and lattice $S_{l}$.
(9)$$S\left(T,B\right)=S_{e}\left(T,B\right)+S_{m}\left(T,B\right)+S_{l}\left(T,B\right).$$

It is essential to mention that the latter equation presumes the separation of the orbital, magnetic, and lattice degrees of freedom. This assumption does not always hold, as evidenced by systems exhibiting Jahn–Teller interactions or magnetoelastic coupling, where these degrees of freedom are intertwined [53,54,55,56].

Magnetic entropy $\left(S_{m}\right)$ strongly depends on the magnetic field. Conversely, it is typically observed that many materials’ electronic and lattice entropies are mainly independent of the magnetic field. However, this is not universally applicable; notably, in low-temperature regimes (approximately below 10 K), some studies suggest that the electronic entropy may exhibit nonlinear dependence on the magnetic field [57].

In this study, we consider a more conventional system where the magnetic field does not influence the lattice and electronic entropies, allowing us to reformulate Equation (Equation 9) as
(10)$$S\left(T,B\right)≃S_{e}\left(T\right)+S_{m}\left(T,B\right)+S_{l}\left(T\right).$$

The quantification of the caloric phenomenon is associated with a thermodynamic refrigerator cycle, where we can take two paths to quantify the effect: (i) an adiabatic trajectory and (ii) an isothermal trajectory. The temperature variation that suffers the systems along the isentropic process is $\Delta T_{ad}$ and is given by
(11)$$\Delta T_{ad}=-\int_{B_{i}}^{B_{f}}\frac{T}{C_{B}}{\left(\frac{\partial S}{\partial B}\right)}_{T}dB,$$
where we use $C_{B}={\left(\frac{\partial U}{\partial T}\right)}_{B}=T{\left(\frac{\partial S}{\partial T}\right)}_{B}$ that corresponds to specific heat at constant *B*.

In the case of quantifying the effect employing an isothermal trajectory, we obtain an entropy variation at constant temperature, ΔS, given by
(12)$$\Delta S=\int_{B_{1}}^{B_{2}}{\left(\frac{\partial S}{\partial B}\right)}_{T}dB.$$

The quantity ${\left(\frac{\partial S}{\partial B}\right)}_{T}$ that appears in the last two equations can be rewritten in terms of the magnetization of the system via Maxwell’s relationship ${\left(\frac{\partial S}{\partial B}\right)}_{T}={\left(\frac{\partial M}{\partial T}\right)}_{B}$.

If we look at Equations (Equation 11) and (Equation 12), we can find a relationship between these quantities. It is found that $-\Delta S\propto \Delta T_{ad}$. Consequently, it is essential to note that when one has a case in which −ΔS>0, we call this kind of response a direct type. The system will heat up, while when the response is −ΔS<0 type, we call this an inverse-type response, and consequently, the system will cool down. Therefore, we expect in a direct response a $\Delta T_{ad}>0$, and for the case of an inverse response, we expect a $\Delta T_{ad}<0$ for the final result of the caloric phenomena.

For the case of the entropy variation at constant temperature *T*, the magnetocaloric expression can be given by the difference in the entropy at the initial and final point of the process as
(13)$$-\Delta S≃-\Delta S_{m}≃S_{m}\left(T,B_{0}\right)-S_{m}\left(T,B\right),$$
where the contributions of $S_{l}$ and $S_{e}$ seen in Equation (Equation 10) vanish due to their dependence only on temperature, and consequently, in an isothermal process (as the one we analyze here), they do not have associated variations as *B* changes. Equation (Equation 13) generates a graph of −ΔS as a function of *T* for a given $B_{0}$ and final *B* that quantifies the magnetocaloric phenomena.

For the temperature and magnetic field ranges we use in our calculations, we fix the *zero* value of temperature at T=0.01, while the smallest field we work is B=0.0. On the other hand, the maximum temperature corresponds to T=4.02, and the magnetic field of B=1.0, all in J units.

## 6. Results

### 6.1. Direct or Indirect Caloric Response?

To understand the caloric response of the system, let us start the analysis with a simple example for Q=2, the top left panel in Figure 1b, and lattices increasing in size from 8×8 to 128×128. The entropy difference that quantifies the MCE in Equation (Equation 13) is given by −ΔS=S(T,0)−S(T,B), where we have selected $B_{0}=0$. This entropy difference is plotted for different final magnetic fields of B=0.2 and B=1.0 as a function of the temperature *T* at which the process occurs, as shown in Figure 6a. We note that the response of the effect is positive for both presented cases of different *B*, which indicates a direct-type behavior for the MCE. It is also observed that for a lattice of 128×128, and below T∼1.3, the thermal response of the effect for B=0.2 and B=1.0 is the same. The latter also occurs for smaller lattices but for lower temperatures than 1.3.

As the temperature increases after T∼1.3, it is observed that the case for B=1.0 is notably different from the one with B=0.2, as the peaks for B=1.0 are larger and shifted to higher *T* for all the lattice sizes. The explanation of this phenomenon is relatively simple. The system prefers to be in a ferromagnetic configuration (i.e., in a single configuration state) in an external magnetic field up until it reaches a maximum in −ΔS. Removing the spin ordering from this state requires a considerable temperature increase in energy. This is reflected in the entropy for Q=2 and a 128×128 lattice for B=0 and B=1, where both entropies are equal up to T∼1.3, and then they separate as *T* increases, see the two lower curves in Figure 5c.

Using the same criteria previously discussed, we will obtain similar behavior associated with the sign of −ΔS for different *Q* values. That is to say, we will obtain a response of the direct type −ΔS>0 for Q=8 even independent of the lattice size, as can be appreciated in Figure 6b. This is because the entropy without an external magnetic field is always greater than the entropy for any non-zero magnetic field value for any value of *Q*, as we can see for the lattice of 128×128 sites in Figure 5c.

In addition, we observe by comparing both panels (a) and (b) of Figure 6 that the maxima for −ΔS/N with 32×32 and 128×128 lattice sites do not present significant variations, either in the magnitude or *T* value where they are located along the horizontal axis. This statement is even more noticeable in the case of Q=8 in Figure 6b, indicating that as the lattice size and *Q* increase, the entropy change curves tend to be similar for the *T* range and *B* values we are considering in these calculations. It is also important to emphasize that an inverse effect −ΔS<0 can be obtained in an isothermal process, starting from a higher magnetic field and going to a lower magnetic field. Still, it is much more natural to have a system without a magnetic field and activate an external field on the spin system afterward.

For the larger *Q* value, we are considering Q=8 in Figure 6b; it is observed that the MCE is more significant for the second transition in all the presented lattice sizes, as seen by the larger maxima in the temperature range from T=1 to T=1.5. This applies to large lattices constructed with Monte Carlo simulations and mean-field theory. However, using an exact formulation performed on the canonical ensemble, the opposite is true for small lattices (e.g., a 3×3 lattice, see Figure A1b for Q=8). Since we are focusing on the thermodynamic limit, we will not delve further into the effect of small lattices in the main text of our work. However, to complement this study, we have added a full Appendix A on exact and mean-field calculations for small systems.

### 6.2. What Q Value Has the Better Magnetocaloric Response?

We have to consider two temperature regimes to answer which value of *Q* maximizes the magnetocaloric effect. Since, in our model, the temperature is in units of J, it is convenient to think of two temperature regimes, the first one for 0<T<1 and the second one for T>1.

From Figure 7, we observe the magnetocaloric response for the 128×128 lattice with even *Q* values. Due to the double-phase transition in the model for Q>5, both Q=6 and Q=8 (circle symbols) display two distinct maxima in the caloric effect as a function of *T*. The first maximum (from left to right in *T*) is associated with an FP-BKT phase transition and the second maxima with a BKT-PP transition. For both Q=6 and Q=8, at temperatures T<1, the magnitudes of −ΔS are nearly the same, but the first maximum occurs for Q=8, reaching its peak at a lower temperature than Q=6. This trend aligns with the expected behavior of our model, where increasing the spin degrees of freedom *Q* allows for similar caloric responses at progressively lower temperatures. As *Q* increases, the temperature at which the FP-BKT transition occurs shifts leftward, approaching zero in the theoretical limit as *Q* tends to infinity. In the higher temperature region T>1, we observe that the caloric effect for Q=8 slightly surpasses that of Q=6, particularly near the BKT-PP transition. This suggests that, for Q>5, the BKT-PP transition becomes more pronounced with increasing spin degrees of freedom *Q*, amplifying the caloric response as *Q* grows.

If we only focus on the maximum magnitude of the MCE and not its location in temperature, this is given for Q=4 for any lattice size. This is due to the sharp peak observed in the specific heat for Q=4 without an external magnetic field, as seen in Figure 4a. If we compare this peak with the one obtained when B=1.0 (Figure 4b), the magnitude peak decreases almost by half, the most remarkable decrease in all the peaks observed for all the *Q* studied. This will generate significant entropic differences and a higher associated caloric effect, as shown in Figure 7. Opposite is the case for Q=2, which has the lowest caloric effect, with its maximum at a higher temperature, thus being the least advantageous for the caloric phenomenology.

As a final comment, we need to clarify the role of the Berezinskii–Kosterlitz–Thouless (BKT) phase in our study. For Q≥5, the system exhibits an intermediate phase characterized by the quasi-long-range order, where spin correlations decay algebraically due to the presence of vortex–antivortex pairs. In a zero magnetic field, this phase is associated with zero magnetization, as the order parameter is non-local. However, in the presence of a non-zero external magnetic field, the Zeeman term in the Hamiltonian breaks the symmetry and induces a finite magnetization, even in the BKT phase. This external field modifies the behavior of the BKT transition, while retaining its essential features, such as power-law spin correlations. The appearance of a double peak in the specific heat for Q≥5 highlights this phase transition, leading to a “shoulder” in the entropy that enhances the magnetocaloric effect near the temperature of the first specific heat peak.

## 7. Conclusions

This study investigated the magnetocaloric effect (MCE) in a *Q*-state clock model on a square lattice using Monte Carlo simulations, focusing on spin systems with varying *Q* orientations. By examining *Q* values from 2 to 8 across different lattice sizes under free boundary conditions, we quantified the MCE through isothermal changes in the external magnetic field, using system entropy as a key metric.

Our results demonstrate that the magnetocaloric effect can be significantly enhanced through the precise control of the spin degrees of freedom. Specifically, for Q≥5, the system undergoes a double-phase transition characterized by two distinct peaks in specific heat, which correlate with an enhanced caloric response. At lower temperatures (T<1), systems with Q=6 and Q=8 exhibit a notable caloric effect driven by the ferromagnetic (FP)-to-Berezinskii–Kosterlitz–Thouless (BKT) transition. Although this transition occurs at different temperatures for each *Q* value, the resulting caloric response remains nearly identical, indicating that increasing the spin orientations shifts the transition temperature while preserving the effect’s magnitude.

The behavior observed for Q=4 is particularly noteworthy, as it presents a single, pronounced peak in the specific heat, maximizing the MCE at a single critical temperature. Unlike higher *Q* systems, which involve more complex double-phase transitions, the Q=4 case offers an efficient entropy change with simplified thermal control, suggesting its potential utility in applications where streamlined caloric responses are advantageous.

In summary, this study provides a deeper understanding of how spin orientations within the *Q*-state clock model influence the magnetocaloric effect (MCE), offering critical insights for optimizing caloric responses in magnetic refrigeration and other thermodynamic systems. These findings not only advance the theoretical comprehension of phase transitions in multi-state systems but also reveal practical strategies for harnessing these phenomena across diverse domains of statistical physics, such as neural networks, biological systems, and percolation theory. The framework established here opens new pathways for future research into optimizing thermodynamic efficiency in spin-based materials, with promising implications for a range of applied technologies.

## Figures and Tables

**Figure 1 entropy-27-00011-f001:**
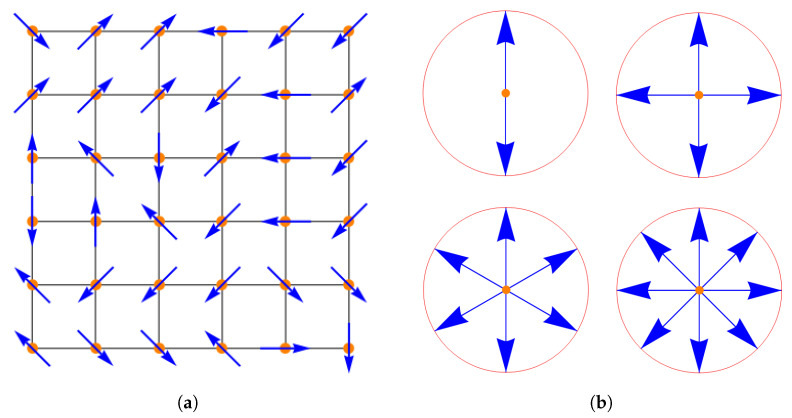
(**a**) Schematic representation for a square lattice of size 6×6 with spin orientations corresponding to Q=8. (**b**) *Q*-clock model for Q=2,4,6, and 8 states. Orange dots indicate sites, and blue arrows are the possible spin orientation at each site with spin vector $\vec{S_{i}}$.

**Figure 2 entropy-27-00011-f002:**
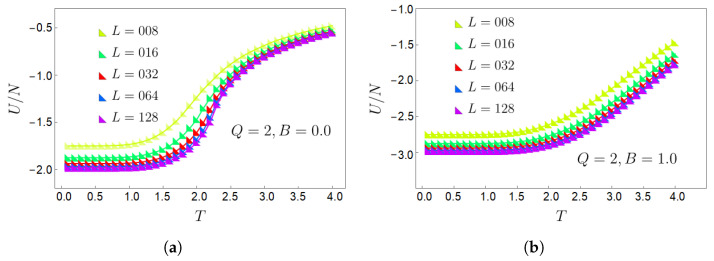
Normalized internal energy U/N as a function of temperature for Q=2 and lattice sizes 8×8, 16×16, 32×32, 64×64, and 128×128 for an external magnetic field of (**a**) B=0.0 and (**b**) B=1.0.

**Figure 3 entropy-27-00011-f003:**
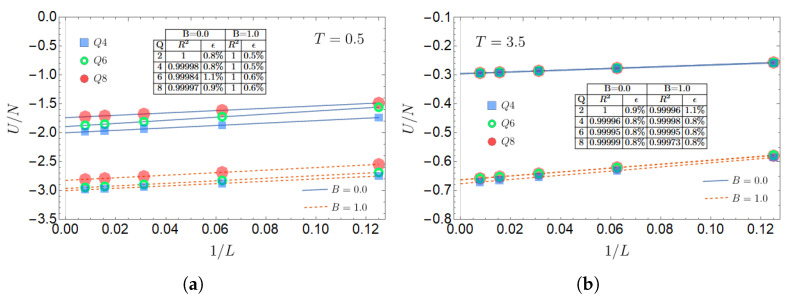
Normalized internal energy U/N as a function of the inverse of the lattice size 1/L, for different *Q* values, namely, 4, 6, and 8, and two values of the magnetic field B=0.0 and B=1.0. The left panel (**a**) corresponds to a low-temperature value, T=0.5, while the right panel (**b**) corresponds to a high-temperature value of 3.5.

**Figure 4 entropy-27-00011-f004:**
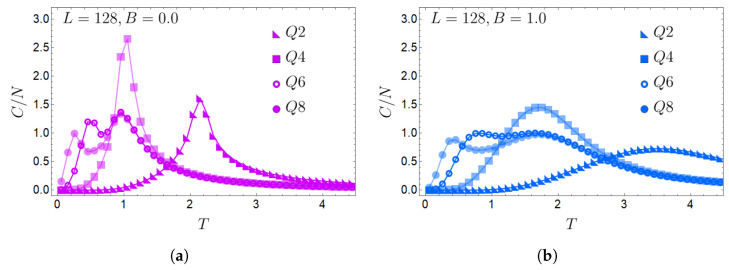
Normalized specific heat C/N as a function of temperature for even *Q* values between Q=2 and Q=8 and a 128×128 lattice. (**a**) B=0.0; (**b**) B=1.0. The shift of the peaks to the right at B=1.0 is clearly seen for all *Q*.

**Figure 5 entropy-27-00011-f005:**
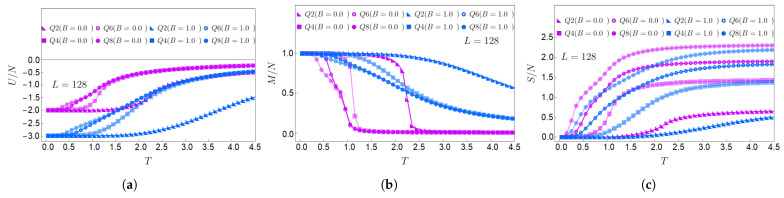
Normalized (**a**) internal energy, (**b**) magnetization, and (**c**) entropy as a function of temperature for even values of *Q* between Q=2 and Q=8 and a 128×128 lattice. B=0.0 (purple curves); B=1.0 (blue curves).

**Figure 6 entropy-27-00011-f006:**
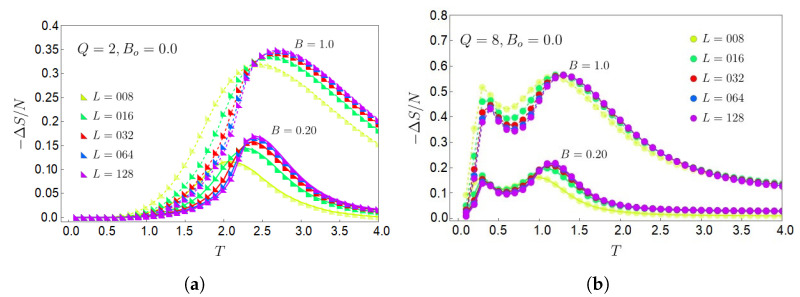
Normalized entropy difference $-\Delta S=S\left(T,B_{0}=0\right)-S\left(T,B\right)$ as a function of temperature for lattices sizes from 8×8 to 128×128 sites, and B=0.2 (lower curves) and B=1.0 (upper curves). (**a**) Q=2; (**b**) Q=8.

**Figure 7 entropy-27-00011-f007:**
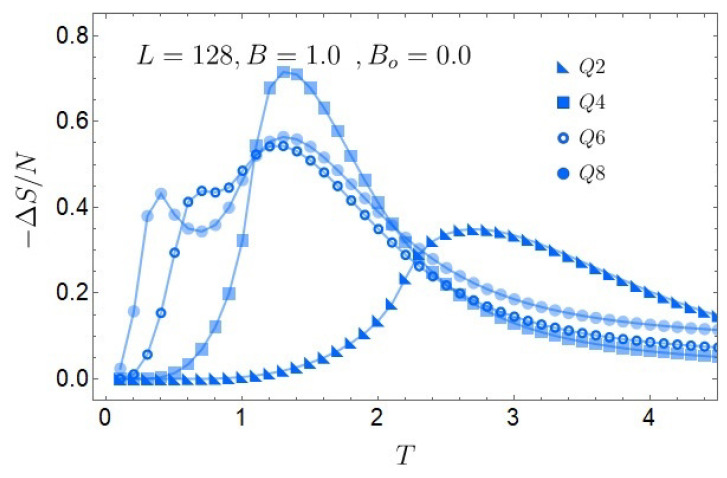
Normalized magnetocaloric effect for a 128×128 lattice using Monte Carlo simulations for the case of Q=2 up to Q=8 and external B=1.0.

## Data Availability

The original contributions presented in this study are included in the article. Further inquiries can be directed to the corresponding author.

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
