# Peer review of "Magnetocaloric Effect for a Q-Clock-Type System"

_entropy, 2024, doi:10.3390/e27010011_

Round 1
Reviewer 1 Report
Comments and Suggestions for Authors
This is a very nice presentation of a topical problem. However there are certain points that deserve amendment. These are the following:
(1) There have been many numerical studies of the magnetocaloric effect, already, so it would be useful to stress more forcefully, just how this study stands out from the previous ones: What does it achieve; that the previous ones didn't address?
(2) What's the significance of the choice of the maximal value of the magnetic field? Is it simply a question of computational cost-and how is it possible to understand this? Does something particular happen at B≥1?
(3) There'e mention of the BKT transition, but this is left quite vague, it would be useful to provide sharp statements, if the BKT transition is, indeed, relevant, or not. One might imagine that it may start to become relevant, in practice, for the higher rather than the lower values of Q. However the statement about a ferromagnetic BKT transition sounds strange, because, across the BKT transition the magnetization does not change-it remains zero, the order parameter isn't a local function of the spin degrees of freedom. So this point requires clarification.
(3) It might be thought that the lattice size shouldn't matter, the computations are, of course, done at finite lattice sizes, however it is the extrapolation to the scaling limit that is relevant. Hence the claim about an optimal lattice size seems disconcerting, since this would imply a non-trivial dependence of ``physical'' parameters on ``laattice'' properties, not just on the size. That's why a discussion of the scaling behavior is required.
In conclusion, the submission does deserve publication, under the condition that the above issues are resolved.
Author Response
Dear Referee, please see attached answers,
Best regards,
Francisco J. Peña on behalf of all the authors.

Reviewer 2 Report
Comments and Suggestions for Authors
This work investigates the magnetocaloric effect (MCE) in a working substance corresponding to a square lattice of spins with Q possible orientations, known as the “Q-state clock model".
The reviewer has the following comments:
1. The manuscript has minor errors, such as line 76 on page two.
2. The reviewer wonders whether the authors can propose some typical materials according to your model, which will facilitate the magnetic refrigeration technique.
Author Response

(The authors gave the same response as above.)

Round 2
Reviewer 1 Report
Comments and Suggestions for Authors
I thank the authors for taking into account the suggestions. The submission is now ready for publication.